# Content validation of an activity-based therapy tracking tool in a community setting for people living with spinal cord injury or disease using cognitive debriefing interviews

Anita Kaiser[1,2,3,4], Hanan Idd[2], Katherine Chan[2], Alexander Djuric[2,4], Sandi Marshall[5], Heather Cairns-Mills[6], Jennifer Leo[7], Kristin E. Musselman[1,2,4]*

1 Rehabilitation Sciences Institute, Temerty Faculty of Medicine, University of Toronto, Toronto, Ontario, Canada, 2 KITE Research institute, Toronto Rehabilitation Institute-University Health Network, Toronto, Ontario, Canada, 3 Canadian Spinal Research Organization, Richmond Hill, Ontario, Canada, 4 Department of Physical Therapy, Temerty Faculty of Medicine, University of Toronto, Toronto, Ontario, Canada, 5 First Steps Wellness Centre, Regina, Saskatchewan, Canada, 6 Walk it Off Spinal Cord Wellness Centre Incorporated, Newmarket, Ontario, Canada, 7 The Steadward Centre for Personal and Physical Achievement, University of Alberta, Edmonton, Alberta, Canada

* kristin.musselman@utoronto.ca

**Data Availability Statement:** Data (i.e., excerpts of transcripts) are provided throughout the paper and

## Abstract

### Background

Activity-based therapy (ABT) has shown promise as a viable therapeutic intervention to promote neurorecovery in people with spinal cord injury/disease (SCI/D). Tools that track the details of ABT sessions may facilitate the collection of data needed to inform best practice guidelines for ABT.

### Objective

The purpose of this study was to evaluate the content validity of a prototype ABT tracking tool.

### Methods

Nine clinicians and five individuals with SCI/D from three community-based ABT clinics in Canada used the prototype tracking tool over three ABT sessions and then participated in individual cognitive debriefing interviews. The interview guide was developed based on recommendations by Brod and colleagues for establishing content validity (i.e., appropriate, comprehensive, comprehensibility). Interviews were transcribed verbatim and analysed using deductive followed by inductive content analysis.

### Results

Overall, the ABT tracking tool was found to have good content validity. Identified categories included: (1) Content validity of the ABT tracking tool. Participants found the tool to be comprehensive and appropriate for all levels and severities of injury. Recommendations to

in the Supporting Information. The full transcripts cannot be shared publicly as per research ethics board guidelines. Data may be available upon request to the University Health Network Research Ethics Board (reb@uhnresearch.ca).

**Funding:** Praxis Spinal Cord Institute.

**Competing interests:** The authors have declared that no competing interests exist.

improve the tool included adding a comment section and additional parameters to each activity. (2) Facilitators of tool use, dissemination and implementation. Using the tool during rest breaks or after the session were suggested to maximize therapy time. Providing the tool as an app and offering education and training on use of the tool were highly recommended. Advertising through community clinics and social media may facilitate dissemination of the tool. (3) Barriers of tool use, dissemination and implementation. The paper format of the tool, added workload, learning curve and challenge to adopt a new documentation system were considered barriers to tool use.

## Conclusions

The prototype ABT tracking tool was validated for content using cognitive debriefing interviews. Recommendations will be used to improve the tool and assist in dissemination and implementation efforts.

## Introduction

Spinal cord injuries and diseases (SCI/D) can have a profound, life-long impact on a person's physical, emotional and psychosocial well-being [1–4]. In the last few decades, significant efforts have been made to develop rehabilitative strategies that facilitate neuroplasticity to increase function and independence. Activity-based therapy (ABT) is a group of interventions that target paralyzed or partially paralyzed areas of the body to promote neurorecovery [5, 6]. ABT is task-specific, intensive and repetitive, however gaps in knowledge exist in our understanding of the appropriate timing to deliver ABT, the method of delivery and the dosage needed to optimize recovery [7–12]. Tools that capture the details of an ABT program, such as the frequency and duration of sessions, the types of activities performed, equipment used, and the challenge level could provide the essential data needed to address these gaps in knowledge. At present, there are two taxonomies that capture the details of a SCI-specific rehabilitation program, the SCIRehab project [13, 14] and the Spinal Cord Injury-Interventions Classification System (SCI-ICS) [15]. Although comprehensive, both taxonomies include disciplines and/or activities that would not be considered ABT, such as social work and activities of daily living, and fail to capture the details that describe dosage, intensity and challenge level, which are fundamental parameters used to characterize ABT [16].

In response to the lack of appropriate tools to document the details of an ABT program or session, our team developed such a tool; a research priority identified by the Canadian ABT Community of Practice (CoP) [17]. To determine what types of ABT to include in a tracking tool, we: (1) identified the characteristics of ABT through a scoping review [16], (2) sought the perspectives of key interest groups on tracking ABT through focus group interviews [18], (3) determined the content to include in an ABT tracking tool through a Delphi survey [19], and (4) triangulated the findings from these three information sources [19]. The triangulated data were used to develop a prototype ABT tracking tool in paper-based form with the intent to eventually develop an app version of the tool. The ABT tracking tool included one technology, transcutaneous neuromuscular electrical stimulation (NMES), and nine types of ABT: treadmill training, overground walking, load-bearing exercise (i.e., standing, 4-point, high kneeling, crawling), transfer training, ergometer training, balance training and other task-specific movement. The most common parameters reported in the scoping review [16] were included as

items to track for each type of ABT and NMES. For example, with ergometer training, individuals could report the duration of the activity, the type of ergometer used (i.e., leg, arm, both, recumbent), the duration of cycling bouts and rest breaks, the cadence, the power output, the number of people assisting, the level of assistance required and the measure of exertion (i.e., Borg rating of perceived exertion, heart rate maximum, blood pressure maximum) (see S1 Text for the prototype ABT tracking tool). Clinicians and people with SCI/D would be able to use the tracking tool to document the details relevant to them for each activity they participate in. This information would provide a record of their sessions and allow clinicians and people with SCI/D to modify the program as appropriate.

As a next step in the development of the ABT tracking tool, this study aimed to evaluate the content validity of the prototype tool using cognitive debriefing interviews to determine whether the tracking tool was comprehensive, appropriate and comprehensible within the context of the construct (i.e., ABT), setting (community clinic) and population (i.e., SCI/D) of interest [20–23].

## Materials and methods

### Study design

A qualitative study was conducted to evaluate the content validity [20, 22, 23] of a prototype ABT tracking tool through semi-structured, cognitive debriefing interviews [20, 21] with individuals with SCI/D and clinicians. Cognitive debriefing is a technique used to assess participants' interpretation of questions or items in a survey or other instrument during an interview [20]. Probing is considered the most suitable technique of cognitive interviewing when assessing the comprehension, comprehensibility and appropriateness of a new instrument [20, 21]. Research ethics approval was obtained from the University Health Network (REB 20–5382). The Standards for Reporting Qualitative Research guided the reporting of study details [24].

### Participants

Three community-based ABT clinics who were members of the Canadian ABT CoP were invited to participate in this study from February 1, 2022 to August 31, 2022. The clinics were located in the provinces of Ontario, Saskatchewan and Alberta. Posters were used to recruit clinicians and individuals with SCI/D from each of the sites. Clinicians had to be a physical therapist, kinesiologist, certified exercise physiologist or occupational therapist, and had to provide ABT to clients with SCI/D for a minimum of once a week for 2 months. Individuals with traumatic or non-traumatic SCI/D were eligible to participate in the study if they: (1) were more than 6 months post-injury or since the onset of neurological symptoms (non-traumatic SCI/D), (2) participated in a community-based ABT program at least once a week for 2 months, and (3) had no other physical, mental or cognitive issues that would preclude them from participating in ABT or this study. Written, informed consent was obtained by a clinical research coordinator by phone, and participants returned a signed copy of the consent form to the research team by email.

Sample size was determined using information power, which considers five criteria: (1) study aim, (2) sample specificity, (3) use of established theory, (4) quality of dialogue and (5) analysis strategy [25]. Generally, high-powered studies (i.e., those with strong methods that capture detailed and specific information from a trained expert) would require a smaller sample size (see Table 1). As such, we aimed for a moderate sample size of 10 to 12 clinicians (2–4 per site) and 10 to 12 individuals with SCI/D (2–4 per site). Among participants with SCI/D, purposive sampling [26] was used to recruit an equal distribution of participants living with

**Table 1. Sample size rationale using information power [25].**

| Key factors | Justification | Favours small or large sample size? |
|---|---|---|
| Study aim: *Narrow or Broad*? | Narrow: The study had a specific aim. | Small |
| Sample specificity: *Dense or Sparse*? | Dense: The sample was specific with well-defined inclusion criteria, but included two participant groups (clinicians and people living with SCI/D). | Moderate |
| Established theory: *Applied or Not*? | Applied: The study did not rely on a theoretical framework, yet did apply recommendations outlined by Brod et al. [20] | Moderate |
| Quality of dialogue: *Strong or Weak*? | Strong: The interviewer had extensive knowledge of ABT and SCI/D and experience conducting qualitative interviews. | Small |
| Analysis strategy: *Case or Cross-Case*? | Cross-case: A cross-case analysis was used to determine the content validity of the ABT tracking tool. | Large |

ABT, activity-based therapy; SCI/D, spinal cord injury or disease.

paraplegia and tetraplegia, with a split of six to seven males and four to five females [27, 28]. Verbal and written consent was obtained prior to participation in the study.

## Data collection

Following enrollment in the study, each participant was provided with a paper-based version of the ABT tracking tool and asked to reflect on the tool over the course of three ABT sessions. During those sessions, participants were asked to engage in their usual activities at the duration and frequency each activity was typically completed. Clinicians had the liberty to reflect on the tool with the same client or different clients for each session, provided the client had a traumatic or non-traumatic SCI/D. Participants with limited hand function could seek assistance from a caregiver or staff member to add reflective notes to the tool if desired.

Once the three ABT sessions were completed, participants were scheduled to complete an individual interview that was held over web conferencing (Zoom Video Communications). Participants were asked to have a copy of the tool as a reference during the interview. Participants were first provided some background information and reminded of the purpose of the study. The interviewer (AK who has over 25 years of experience living with SCI, ten years of experience in qualitative research and eight years of experience participating in ABT) then asked participants a set of demographic questions, which included age, sex, gender and years of experience with ABT. In addition, clinicians were asked about their occupation and years of experience with SCI/D, while participants with SCI/D were asked about their years post-injury, injury mechanism, neurological level of injury and severity of injury.

Following the questionnaire, the interviewer (AK) conducted a cognitive debriefing interview using a semi-structured interview guide. The interview guide was developed based on recommendations by Brod et al. [20] to assess content validity of new or existing tools in qualitative research. The interview guide contained open-ended questions and the verbal probing technique [21] was used to understand participants' perspectives on the content of the tool, the ease of use, and whether it was comprehensive, comprehensible and appropriate for all levels

and severities of SCI/D. The interviewer also sought feedback on the preferred method of delivery (e.g., paper or an app) and whether participants perceived any barriers and facilitators to tool use and implementation into practice (see S2 Text for the interview guide). A team member (KEM, who had extensive experience in SCI/D, ABT and qualitative research or AD who was a novice in all areas) attended each interview and took reflexive notes [29] to capture the main idea, personal biases and any additional details, such as their impression of the emotional state and openness of participants to share their perspective. Team meetings were held to discuss completed interviews and add additional probes to the interview guide. The interviews were audio-recorded and transcribed verbatim by a team member (HI who was a novice in all areas) in Microsoft Word (2021).

### Data analysis

Participant demographic information was summarized in table format using descriptive statistics. A directed (deductive) content analysis [30, 31] was used, where the interview guide was used deductively to derive the initial set of categories. The transcribed interviews then underwent a conventional (inductive) content analysis [30, 31] whereby team members AK and HI initially read through the transcripts to familiarize themselves with the data, then separately coded a transcript and met with team member KEM to discuss the initial coding list and develop a preliminary codebook. Team members AK and HI coded the remaining transcripts separately and added new codes to the codebook as new information arose. Meetings were held weekly with KEM to discuss the codes, group them into subcategories and map them to the pre-determined categories. If any codes or subcategories did not align with the identified categories, a new category was created. An iterative process was used throughout as codes and subcategories were merged and/or grouped together to form the final set of categories.

Microsoft Excel (2021) was used to assist with data management and analysis. Data from individuals with SCI/D and clinicians were analyzed separately to compare across groups. Content validity was evaluated by assessing participants' perspectives on the appropriateness, comprehensiveness and comprehensibility of the ABT tracking tool. Suggested revisions to the ABT tracking tool were charted in a table (see S1 Table). The findings will be used to refine the ABT tracking tool. Trustworthiness [32] was established through the expertise of team members, investigator triangulation, the use of verbatim quotes and reflexivity.

### Results

Fifteen individuals participated in the study; five individuals with SCI/D and ten clinicians. One clinician (female exercise physiologist) dropped out of the study shortly after enrollment due to health reasons. Demographic information of participants is presented in Table 2.

Interviews occurred between March and May 2022 and ranged from 24 to 91 minutes in length. ABT sessions ranged from one to three hours in length with the most common being two hours. All activities within the tool were reflected on by at least five participants (range 5–14), with representation from both clinicians and people with SCI/D. Muscle strengthening and load-bearing in standing were the only activities reflected on by all participants. Some participants, particularly clinicians, were able to reflect on activities that they were familiar with in practice, but had not engaged in during their three ABT sessions. Table 3 presents the number of participants who reflected on each activity by participant group.

Most clinicians reflected on the ABT tracking tool with two different clients, but a couple of clinicians chose to reflect on the tool with multiple clients across several ABT sessions in order to reflect on all activities within the tool. Most participants reflected on the tool during the ABT session or shortly after the session ended. Some clinicians chose to wait until the end of

**Table 2. Participant demographics.**

| Clinicians (n = 9) | |
|---|---|
| Age (mean ±SD, range) in years | 27.8 ±3.9, 22–30 |
| Sex | 2 Males, 7 Females |
| Gender | 2 Men, 7 Women |
| SCI/D experience (mean ±SD, range) in years | 3.8 ±3.1, 1–9 |
| Occupation | 1 Exercise physiologist<br>7 Kinesiologist<br>1 Physical therapist |
| ABT experience (mean ±SD, range) in years | 3.9 ±2.8, 1–9 |
| Person living with SCI/D (n = 5) | |
| Age (mean ±SD, range) in years | 32.4 ±7.8, 25–45 |
| Sex | 2 Males, 3 Females |
| Gender | 2 Men, 3 Women |
| Years post-injury (mean ±SD, range) in years | 9.2 ±5.1, 3–17 |
| Injury mechanism | 2 Fall<br>2 Motor vehicle<br>1 Sport |
| Injury level | 5 Cervical |
| Injury type | 5 Tetraplegia |
| Injury severity | 2 Complete<br>3 Incomplete |
| ABT experience (mean ±SD, range) in years | 8 ±4.8, 2–15 |

n, number; SD, standard deviation; SCI/D, spinal cord injury or disease; ABT, activity-based therapy.

their day and used the tool in conjunction with documentation methods they currently use at their facility, while others chose to fill out the form as though they were using it in practice. One clinician had a student help record information on the tool during the session and one participant with SCI/D whose English was not a first language had their therapist fill out the tool on their behalf and explain some terminology to them.

Overall, participants liked the tool and the consistent way in which it was formatted, with data tables and each type of ABT on its own page. Words commonly used by participants to

**Table 3. Number of participants who reflected on each ABT activity by participant group.**

| ABT activity | SCI/D (n = 5) | Clinician (n = 9) | Total |
|---|---|---|---|
| Treadmill training | 4 | 5 | 9 |
| Overground walking | 2 | 7 | 9 |
| Muscle strengthening | 5 | 9 | 14 |
| Ergometer training | 4 | 5 | 9 |
| Load-bearing exercise in standing | 5 | 9 | 14 |
| Load-bearing exercise in 4-point | 5 | 6 | 11 |
| Load-bearing exercise in high kneeling | 4 | 8 | 12 |
| Load-bearing exercise in crawling | 3 | 5 | 8 |
| Transfer training | 2 | 3 | 5 |
| Balance training | 5 | 5 | 10 |
| Other task-specific movement | 2 | 6 | 8 |

ABT, activity-based therapy; SCI/D, spinal cord injury or disease.

describe the tool included "user-friendly", "straightforward", quick and "easy to use". Participants also thought the information recorded in the tool was "relevant", "beneficial", especially for insurance companies, and a great way to keep a record of sessions and track progress. A couple of participants indicated that although the tool had a good foundation and "captures the big picture", (Clinician 10) they did feel it lacked the detail needed to describe the intricacies and finer details of the activities.

Three categories were identified through deductive analysis from the interview transcripts: (1) Content validity of the ABT tracking tool; (2) Facilitators of tool use, dissemination and implementation; and (3) Barriers of tool use, dissemination and implementation. Categories 1 and 2 were further divided into subcategories through deductive and inductive analyses, respectively.

## 1) Content validity of the ABT tracking tool

Overall, the ABT tracking tool demonstrated good content validity. Participants were asked to reflect on the three aspects of content validity: the tool's appropriateness, comprehensiveness and comprehensibility, as well as provide feedback on ways to improve the ABT tracking tool.

**Appropriateness.** When asked about the appropriateness of the ABT tracking tool, most participants agreed that the tool was suitable for all neurological levels and severities of SCI/D. A few participants acknowledged that not all individuals would make use of all aspects of the tool as they may not participate in all types of ABT listed on the tool due to their functional ability. A couple of participants thought that individuals with greater function would make better use of the tool because they would be able to participate in all types of ABT compared to individuals with less function who would be unable to perform tasks such as overground walking. "The more independent [the] spinal cord injury patient, this [tool] is more useful." (Person with SCI/D 01) In contrast, another participant thought the tool was more useful for individuals with limited function because for a "higher functioning individual. . .I had a lot less to check off. Like everything was minimal or nothing or no help." (Clinician 05)

**Comprehensiveness.** In general, participants felt the tool was comprehensive and captured most of the activities they engaged in, with one participant commenting "pretty darn good list of everything that an activity-based therapy session should include." (Clinician 06) Some participants appreciated the section on 'Other Task-specific Movement' since "there are some different or weird things, exercises that [I] can't really say what they are, what we do, but it's still good to track so that "Other" category is, I think is really beneficial." (Person with SCI/D 03) Most of the other suggestions provided by participants included the addition of new parameters and expansion of existing parameters to track. For example, one participant suggested adding the type of surface for load-bearing in standing and a few participants suggested expanding the options of level of assistance provided by a therapist.

> For the level of assistance, I had it marked on this page, was whether to include. . .verbal cues or tactile cues, 'cause I did find that with my higher functioning individuals they sometimes required a bit more verbal cues, and I still consider that a level of assistance. . .just like a way to kind of mark whether it was verbal or hands-on tactile cues. (Clinician 05)

Several participants described a desire to record certain parameters, such as walking speed and distance, for each bout of exercise within a session as it often fluctuates based on fatigue level and program design (e.g., exercise pyramids). Participants also had suggestions that applied to all activities within the tool, such as adding a comment section to the bottom of each activity page to record any relevant details not captured within the tool. "For each page,

have a little comment notes section for said therapist to fill in. Maybe the little things on, just the little stuff that you want to remember." (Clinician 10) One area where participants were not in agreement with was tracking the measures of exertion (i.e., Borg rating of perceived exertion, heart rate maximum, blood pressure maximum). Some participants admitted to never tracking these parameters and not seeing a need for it unless a person was newly injured. "The Borg rating and the heart rate, I probably wouldn't record [it]." (Person with SCI/D 05) Other participants thought these parameters were only relevant for some activities, such as load-bearing in standing, and a few participants wanted the ability to record measures of exertion for activity bout. "Include a measure of exertion for each interval of walk if you are doing more than one, just so that you can measure fatigue over time as the intervals progress." (Clinician 08) See S1 Table for a list of all suggested revisions by activity type with supporting quotes.

**Comprehensibility.** Participants described a few aspects of the tool that they found unclear. One issue brought up by most participants was how to record exercises that involved more than one type of ABT. Participants, like this individual with SCI/D who performed a muscle strengthening exercise while in standing, were unsure whether to record the information under both sections *Muscle Strengthening* and *Load-bearing Standing*, "I don't know how you'd say you're on the total gym, but you're doing strengthening exercises. Do you fill out a section on load-bearing standing and then also fill out a section on muscle strengthening?" (Person with SCI/D 04) One clinician reasoned that the decision should be based on the goal of the exercise, "I guess reflecting on the goal, the goal of the movement, the goal of the exercise. If we're trying to get specific with his ball kicks then it would fall into the ['Other Task-specific Movement'] category, but in this instance, I would put it under muscle strengthening." (Clinician 04) Some participants were unsure about the difference between the ABT activities 'muscle strengthening' and 'other task-specific movement' and when you would choose to record information under one category versus the other. "It's very general. . .Probably very similar to the muscle strengthening section. I think I'd kind of use one or the other 'cause for me they seem very similar." (Clinician 05) Another aspect of the tool that some participants were uncertain about was the Borg rating of perceived exertion. Some individuals with SCI/D, who may have been unfamiliar with the Borg, struggled with using the scale as this clinician describes:

> Using the Borg scale, it's a tricky one 'cause usually standard is just like, "on a scale of one to ten how was this exercise?" So, most of our clients they like to do an RPE scale of one to 10, but that's just like could be clinic to clinic wise what type of RPE scale they use 'cause out of. . .six to twenty-two is a little bit sometimes hard for people to comprehend. (Clinician 03)

A few participants also described struggling with reporting certain parameters, such as crawling speed if an individual only moved a short distance for that activity or the amount of body weight supported by the individual's upper body if they used assistive devices, like a walker.

## 2) Facilitators of tool use, dissemination and implementation

Participants provided several suggestions to facilitate use and implementation of the tool in practice as well as ways to raise awareness and education about the tool to encourage broader dissemination and implementation.

**Facilitators of tool use in practice.** Participants with SCI/D believed it would be best to record information during their session rest breaks after completing an activity to maximize

therapy time or have their therapist document their session and then add any additional details after the session. Some clinicians agreed with filling out the tool during a session to avoid recall issues, provided a client did not require heavy hands-on assistance. Other clinicians felt that recording data after each client was sufficient and similar to current documentation methods. In addition, these clinicians claimed that once they became familiar with the tool, they would know what they needed to remember. A few clinicians said it would depend on the format of the tool, "If it was a paper copy, I could see myself filling [it] out more so after, but if it was an app, like a quick kind of thing, I could see myself doing it during, like kind of in between exercises, during rests and stuff like that." (Clinician 07)

**Implementation facilitators.** Participants recommended several elements that could be considered to help facilitate implementation of the ABT tracking tool in community-based clinics. A participant with SCI/D thought that having clinicians explain the tool to their clients would be a good way to familiarize people with the tool and gain comfort using it, and a few clinicians suggested hosting a training session, "I think the idea of like a tutorial or informational session would alleviate any issues with learning how to use the tool." (Clinician 07) Participants also thought that having equipment, such as wearable technology that can record measures of exertion and blood pressure, would be helpful during a session and more accurate than subjective measures. Another criterion, reported by a clinician, that would facilitate use of the tool is having additional personnel available to record the information while they are providing ABT to their clients, "We had the luxury of having some extra hands around and students around...I think it's nice to have someone to do that for [you] and then you can kind of keep it, your eye on the client a little bit more." (Clinician 06) On a similar note, nearly every participant suggested that providing the ABT tracking tool in a digital format, such as an app, would facilitate its use. The words they used to describe an app were "better, easier, simple, a lot quicker, accessible and convenient". A couple of clinicians pointed out that an app would easily be streamlined into use since it would complement their current routine of documentation, "All of our documenting is done electronically anyways, so...personally and honestly the best way for me to comply to this is just to be able to tack it onto what I'm documenting." (Clinician 03)

**Promoting dissemination and implementation through awareness and education.** Regarding the best ways to make people aware of the tracking tool, participants suggested sharing the tool with community clinics and rehabilitation facilities that are providing ABT. "I think contacting the facilities that offer activity-based therapy and letting them know and then they can let their clients know." (Person with SCI/D 05) As one clinician described, this may involve "reaching out to the people on the ground, like trying to get the message out to specific clinics, maybe talk to admin there and get the word out to the clinicians." (Clinician 08) From there, participants believed that once clinicians started using the tool, seeing the benefits of it and promoting it, then their clients would follow suit. "It has to be the trainers using it and showing how great it is...If the trainers are using it and tracking their progress on it...that's the selling point." (Clinician 04) Several clinicians also proposed showcasing the tool at conferences and forums where delegates would have an opportunity to try out the tool.

> There's a couple spinal cord injury conferences...I would assume that probably the spinal cord injury conference would probably be the best way to get a little bit of time in there and just say like "hey, this is what we're doing. This is why we're doing it. If you're interested in it, if you want to be a part of it, this is your opportunity to sign up" or create a relationship or whatever it might be. (Clinician 06)

Another option recommended by a couple of participants was introducing the tool into the educational curriculum for students in the health profession to learn about it and share it on

their work placements. "I know that some universities are also starting to offer an elective ABT course, which I think is gonna [be] amazing for the field." (Clinician 03) However, the most popular advice given by participants was to share the tool broadly through social media networks and allow it to spread through 'word of mouth'.

> I think social media is a big thing. Like getting it out there and advertising it that way. But also just. . .clinics knowing about it, and then them telling their clients also, or like other clinics that they know. . .I think it'd be the best way to [go by] word of mouth. (Person with SCI/D 02)

### 3) Barriers of tool use, dissemination and implementation

Participants described a few issues that they felt might hinder use and implementation of the ABT tracking tool in practice. One key challenge was the paper format of the tool. A couple of participants with SCI/D admitted to not having the functional ability to record their information on the form. "I can't write so if I had to write on the paper, that would be an issue." (Person with SCI/D 05) Clinicians also had concerns with the volume of paperwork they would have to manage during and after a session and the difficulty with reviewing and analyzing that data afterwards. Some participants, like this individual with SCI/D, thought the tool was time-consuming to fill out, "If [it] takes too long to record sessions, people are going to be less likely to do it." (Person with SCI/D 04) A few participants also admitted feeling overwhelmed with the tool when they first used it and conceded that there was a learning curve to get comfortable with using the tool. "It could seem overwhelming right off the start, so I don't know if that would deter people away." (Person with SCI/D 03) Clinicians specifically acknowledged the difficulty with changing over to a new system and familiarizing themselves with a new tool, particularly when they were already documenting their sessions through the use of SOAP notes or on a notepad. There was also the notion of having one more item added to their workload.

> I think it would be like any new tracking tool or anything. Getting people to have one more type of charting, as well as the unfamiliarity with it in the beginning and if you're not in the habit of using it quite yet, it can seem daunting or just like an extra task that you may not have time for. . .would be the kind of challenges I would see being faced. (Clinician 05)

In addition, clinicians recognized that individuals with SCI/D would likely not use the tool if their therapists were not already using it, "If the trainers aren't using it, it'd be really hard to get the clients to want to record all these things every single night." (Clinician 04).

## Discussion

This study evaluated the content validity for a prototype ABT tracking tool through cognitive debriefing interviews with clinicians and people with SCI/D who participate in ABT in a community-based setting. Overall, the tool has content validity and was deemed appropriate for individuals with SCI/D of all injury profiles. Participants considered the tool to be comprehensive in capturing the various types of ABT, however, they did provide recommendations to increase the level of detail, and improve the comprehensibility and ease of use of the tool. The paper-based format of the tool was considered to be a barrier to use and implementation into practice, and there was widespread agreement to offer this tool as an app to facilitate use. Offering education and training on how to use the tool, as well as promoting the tool through

community-based clinics and social media platforms were recommended strategies to facilitate dissemination and implementation of the tool.

Participants in this study provided a number of valuable suggestions to improve the comprehensiveness and comprehensibility of the tool. These suggestions are important not only for supporting the content validity of the tool, but also since existing taxonomies [13–15] lack details that are characteristic of ABT. For example, intensity (i.e., achieving a moderate to high cardiovascular load) is a defining characteristic of ABT and should be monitored throughout an ABT session. Although participants disagreed on the need to track measures of exertion (i.e., Borg rating of perceived exertion, heart rate maximum, blood pressure maximum), we believe it is important to track these measures across all activities. Intensity is one of the key details missing from pre-existing rehabilitation taxonomies [13–15] so its inclusion makes our tool more appropriate as a method of documentation in ABT programs. Another important feature of the ABT tracking tool is the level of detail requested about the dosage (e.g., number of sets, repetitions, bouts) and challenge level (e.g., walking speed, cadence, amount of resistance). While both the SCI-ICS and the SCIRehab project track the time spent on an activity, and the SCIRehab project tracks some parameters (e.g., the amount of assistance required to perform an activity, total distance walked, number of steps), neither tool offers a comprehensive set of parameters to effectively track dosage, which is necessary to continuously assess and monitor challenge level to inform program modification and aid in progression. Table 4 presents a summary of the modifications that have been made to the ABT Tracking Tool based on the feedback received from participants. Decisions were partially based on preserving data quality and ensuring feasibility of aggregating data for meaningful interpretation; a challenge in big data analytics [33, 34]. The modifications will be incorporated into the updated version of the prototype ABT tracking tool.

An area of confusion for some participants was knowing where to document activities that included more than one type of ABT, such as balancing and load-bearing in standing. As recommended by a participant who is a clinician, in these situations, a review of the activity's goals should be considered and used to determine which type of ABT is the primary focus of the activity. The additional type(s) of ABT involved in the activity can be noted under 'Additional levels of difficulty' as appropriate. Previous literature describes the importance of goal setting in ABT, with programs customized and incorporating combinations of activities to challenge individuals to meet their goals [8, 38].

Most participants in this study thought the ABT tracking tool was appropriate for all levels and severities of SCI/D, although a few individuals noted that individuals who were very high or very low functioning may not make full use of the tool. This is not a concern as the intent of the tool was not to complete every detail for every type of ABT activity. Arguably this is exactly the type of information we would want to collect in order to understand if and how individuals with differing levels of SCI/D severity participate in the various types of ABT activities. Existing taxonomies, which are non-ABT focused, were found to be suitable for all levels and severities of injury, yet acknowledged that some activities only apply to specific sub-groups of SCI/D [13, 39]. A study by Quel de Oliveira [40] looking at the effects of multi-modal ABT in a community-based setting included individuals of all levels, severities and time post-injury. Similarly, two reviews on ABT described participants across included studies with a broad range of personal demographics and injury characteristics [16, 41]. This suggests that an ABT tracking tool which includes all relevant types of ABT should be suitable for all individuals with SCI/D.

A critical barrier of the ABT tracking tool, reported by participants in this study, was the availability of the tool in paper-based form. Although the original intent was never to provide a final version of this tool in paper-based form, it was encouraging to see widespread agreement among participants about the benefits and utility of providing this tool as a mobile app.

**Table 4. Modifications to the ABT tracking tool based on participant feedback.**

| Type of ABT | Suggested Revision | Decision |
|---|---|---|
| All | • Add a comment box | Accept |
| | • Add 'Goal of exercise' | Accept |
| | • Add 'Technology and/or equipment used' | Accept |
| | • Include 'Measures of exertion' for each bout | Accept |
| | • Expand parameter 'Overall level of assistance' to include 'Full assistance, Full independence, Spotting and Verbal/Tactile cues' | Accept with modification—Change heading to 'Overall level of physical assistance'. Verbal cues are inherent of ABT, 'Total assist' is already an option and anyone who is fully independent would skip this parameter. Any additional details may be added to the comment section. |
| | • Add 'Talk test' to 'Measures of exertion' | Accept with modification–Add 'Talk test' as a measure of exertion with 3 stages as options [35–37]: <br> Stage I: Able to speak/sing comfortably (low intensity; <50% HRmax) <br> Stage II: Able to speak with mild difficulty; unable to sing (moderate intensity, 50–75% HRmax) <br> Stage III: Able to speak with extreme difficulty (high intensity; >75% HRmax) |
| | • Add check box for 'Duration' if they reached fatigue limit | Accept with modification—Add the following text to 'Measure of exertion': 'Activity stopped due to muscle fatigue'. |
| | • Add parameter 'Assistance from equipment' (i.e., safety set-up, support, assist) and open space to indicate type of set-up and level of assistance provided by the equipment | Decline–Difficult to accurately interpret the amount of support provided by equipment. 'Technology and/or equipment used' and 'Overall level of physical assistance' provided by clinician are recorded. Any additional details may be added to the comment section. |
| | • Add 'Level of assistance' for each body part | Decline–May be time consuming to record. 'Overall level of physical assistance' and 'Technology and/or equipment used' will capture this information. Any additional details may be added to the comment section. |
| | • Add 'Other' as a new page | Decline–The tool is only tracking ABT. Other types of ABT that do not fit under any activity can be recorded on the sheet 'Other Task-specific Movement'. |
| | • Add 'Cardiovascular exercise' as a new page | Decline–Not all exercise is ABT and cardiovascular load is embedded in ABT. |
| | • Add 'Reflex integration/nervous system recruitment' as a new page or component of each activity | Decline–All ABT involves nervous system recruitment and reflex integration. Any additional details may be added to the comment section. |
| | • Remove 'Measures of exertion' from all activities | Decline–As ABT aim to achieve a moderate to vigorous cardiovascular workload, there should be an option to record measures of exertion for all activities. |
| | • Keep 'Measures of exertion' for only some activities (i.e., loadbearing exercise in standing and overground walking) | Decline–As ABT aim to achieve a moderate to vigorous cardiovascular workload, there should be an option to record measures of exertion for all activities. |
| Treadmill Training | • Add 'Walking speed' for each bout | Accept |
| | • Add 'Level of Difficulty' for each bout | Accept |
| Overground Walking | • Add 'Walking distance' for each bout | Accept |
| | • Delete unit of measure for 'Duration of activity' | Decline–Standardizing units of measure will preserve data quality and simplify data aggregation. |
| | • Add other units of measure for 'Walking speed' | Decline–Standardizing units of measure will preserve data quality and simplify data aggregation. |
| | • Add 'Number of steps' for 'Walking distance' | Decline–Standardizing units of measure will preserve data quality and simplify data aggregation |

(*Continued*)

**Table 4.** (Continued)

| Type of ABT | Suggested Revision | Decision |
|---|---|---|
| Muscle Strengthening | • Add option 'Whole body' for 'Muscles targeted' | Accept |
| | • Add FES parameters for each muscle group | Accept |
| | • Add 'Hold duration' for each bout | Accept with modification–Standardize units of measure (i.e., seconds). |
| | • Add 'Other' for external load and 'Therapist resistance' | Accept with modification–Add 'Therapist resistance' as an option. Any additional details (e.g., theraband colour) may be added to the comment section. |
| | • Add 'Position' and different surface options | Accept with modification–Add 'Position' with the following options: lying on back, lying on stomach, sitting, long sitting, standing. Surface options may be captured in open comment box. |
| | • Add option to track 'Palpable contraction/participation' or 'Initiation of spasm/tone' | Decline–This information will be reflected by tracking 'Overall level of physical assistance'. Any additional details, such as the occurrence of spasms, may be added to the comment section. |
| Ergometer Training | • Add 'Standing and Kneeling positions' | Accept |
| | • Add 'Cadence' for each bout | Accept |
| | • Add 'Number of revolutions' for each bout | Accept |
| | • Add options for resting position (e.g., in sitting or standing) | Decline–Any additional details may be added to the comment section. |
| Loadbearing– Standing | • Add 'Type of surface' | Accept with modification–Add the following surface options: stable and unstable. Any additional details may be added to the comment section. |
| | • Add 'Level of difficulty' for each bout (e.g., none/single/double arm, muscle strengthening, ball throw, weight shifting, mini squats, knee bends, single leg, planking, balance, reaching) | Accept with modification–There are many ways in which standing can be challenged. Add 'Level of difficulty' for each bout with open text for the specific challenge to be recorded. |
| | • Add 'Vibration' and parameters | Decline–Vibration did not meet the criteria for inclusion in an ABT tracking tool in our prior research [19] |
| Loadbearing– 4-Point | • Add 'Level of difficulty' for each bout (e.g., perturbation, ball tap, cat and dog, balance, push-ups, muscle strengthening, 3-point, burpee) | Accept with modification–There are many ways in which 4-point loadbearing can be challenged. Add 'Level of difficulty' for each bout with open text for the specific challenge to be recorded. |
| Loadbearing–High kneeling | • Add position 'low kneel (i.e., hero pose) and high kneel' for each bout | Accept |
| | • Add 'Level of difficulty' options for each bout (e.g., muscle strengthening, ball throw/hit, single arm, balancing, perturbations, reaching) | Accept with modification–There are many ways in which low and high kneeling can be challenged. Add 'Level of difficulty' for each bout with open text for the specific challenge to be recorded. |
| | • Add 'Support overhead' and 'Unstable/Stable surface' | Decline—'Technology and/or equipment used' will capture this information. Any additional details may be added to the comment section. |
| Loadbearing– Crawling | • Add 'Other' for distance unit of measure for each bout | Accept with modification–Unit of measure will remain the same to preserve data quality and simplify data aggregation, but distance will be added for each bout. |
| | • Add 'Harness' to body weight support | Decline–'Technology and/or equipment used' will capture this information. |
| Transfer training | • Add 'Ending seat height' and 'Other' for seat height | Accept with modification–Add 'Ending seat height' in centimeters or inches |
| | • Add 'Type of transfer' as a parameter (e.g., stand and pivot, sit /stand, sit/sit, hero/high kneel, plank/4-point) | Decline–Information will be captured by start/end position and seat height (if relevant). Any additional details may be added to the comment section. |
| | • Add 'Level of difficulty' (e.g., hard or soft surface) | Decline–Information already captured under 'Starting position/surface of transfer'. |
| Balance Training | • Add 'Static or Dynamic' options | Accept with modification–Add 'Static, Dynamic and Reactive' as options. |
| | • Split 'Sitting' into 'Short and Long sitting' | Accept |
| | • Add 'Level of difficulty' for each bout (e.g., internal/external perturbation, ball throw/catch, one leg) | Accept with modification–There are many ways in which balance control can be challenged. Add ''Level of difficulty' for each bout with open text for the specific challenge to be recorded. |

(*Continued*)

**Table 4.** (Continued)

| Type of ABT | Suggested Revision | Decision |
|---|---|---|
| Other Task- specific Movements | • Add 'Duration' for each bout | Accept with modification–Add standardized unit of measure (i.e., minutes) |
| | • Add 'Intensity' for each bout with unit of measure open | Decline–'Measures of exertion' and 'Number of Repetitions' included and reflect appropriate measures of intensity. Any additional details may be added to the comment section. |

ABT, activity-based therapy; HRmax, maximum heart rate; FES, functional electrical stimulation.

These sentiments were iterated in a previous study that explored perspectives on tracking ABT from multiple key interest groups [18]. The app would enable suggested changes expressed by participants in this study to be easily incorporated into the current version of the tool. Most importantly, an app for ABT would have the potential to feasibly and easily aggregate data across participants and clinic sites to inform the development of clinical guidelines and shed light on the types of ABT exercises and dosage required to optimize recovery. The use of medical software applications in clinical practice has seen enormous growth in recent years, with the driving force being the need for tools at the point of care [42]. Mobile apps in healthcare may be used for a multitude of tasks including information gathering, monitoring, clinical decision-making and evidence-based education [42, 43]. The benefits of an app reported in this study have been described in other work such as convenience, increased efficiency, improved accuracy, better clinical decision-making and enhanced productivity [42]. Currently, most health apps developed for the SCI/D population are focused on self-management or self-care and monitoring, such as bowel and bladder management and skin monitoring (e.g., iMHere, SCI Health Storyline) [43]. A prototype app in the US (SCI-Ex) has been developed to promote fitness in the SCI/D population by educating them on adaptive exercise through video demonstrations and enabling them to set and track their fitness goals and exercise routine [44]. Although discontinued in 2019 pending further upgrades, the training videos are still available and may provide a useful example when developing the ABT app or modifying an existing app if deemed suitable.

Education and training were mentioned as important facilitators to tool use. Toolkits have been increasingly adopted into healthcare practice as a method to support knowledge translation and implementation of evidence into practice [45]. Salbach and colleagues [46] report on the iWalk toolkit developed to guide physical therapists in using the 10-Metre Walk Test and 6-Minute Walk Test in the stroke population. An evaluation of the toolkit indicated a significant difference in uptake of test usage post-intervention [47]. A systematic review that evaluated the effectiveness of five toolkits created for individuals with distinct chronic conditions (cancer, cardiovascular disease, diabetes) to encourage adherence to physical activity found positive trends towards an increase in physical activity with some toolkits. The toolkits varied in their conceptual frameworks, such as the use of behaviour theory, and evidence-based content which may have influenced outcomes [48]. A toolkit to support the use and implementation of the ABT tracking tool could contain a collection of resources for both clinicians and individuals with SCI/D who already deliver/participate in ABT or newcomers. This may include resources that could provide users with knowledge and understanding about ABT principles and practice, along with definitions of key terms. Information may be provided through a combination of text, illustration and video demonstration. This would lay a foundation for understanding the purpose and benefits of an ABT tracking tool. A simple instruction manual that explains how to use the tool could be included to enable effective use of the tool and allow for greater consistency and reliability in documentation.

Ultimately, the intent of the ABT tracking tool is to be used by clinicians and people living with SCI/D to record session details that are important and applicable to them. Maintaining consistency is the key to procure meaningful data. Aggregate data from multiple individuals with SCI/D across multiple clinic sites would provide the information needed to detect trends in equipment use and types of ABT practiced. Combined with data concerning ABT outcomes, tracking data from the tool (e.g., types of ABT, dosage, exercise intensity) may be used to answer important questions such as the optimal timing, methods and dosage to maximize recovery, as well as support the development of best practice guidelines. The utility of big data in transforming healthcare and improving health outcomes is becoming widely recognized. The collection, storage and analysis of large amounts of data into meaningful pieces of information can aid in the decision-making process for the prediction and diagnosis of disease conditions and evaluation of treatment efficacy [34, 49]. A systematic review highlighted numerous examples, such as big data analytics being used to help predict complications (e.g., hypoglycemia) in individuals with diabetes or to diagnose mental health disorders [33]. In Canada, the Standing and Walking Assessment Tool (SWAT) has been included in the Rick Hansen SCI/D Registry (RHSCIR) and rehabilitation centres of the Spinal Cord Injury-Implementation and Evaluation Quality Care Consortium (SCI-IEQCC). Data collected from the SWAT has indicated that more time spent on gait and pre-gait activities translated to a greater improvement in SWAT staging and may provide guidance in future on the optimal timing and intensity of rehabilitation [50]. Next steps will be to revise the prototype ABT tracking tool and convert it into an app or modify a pre-existing app for further feedback and psychometric tests.

## Limitations

Participant recruitment and data collection occurred during the COVID-19 pandemic. During this time period, the three-participating community-based ABT clinics were closed or operating below full capacity for periods of time. The clinics also reported a change in client demographics with fewer individuals living with SCI/D attending ABT sessions. This had a significant impact on the recruitment of individuals living with SCI/D in this study. However, even though only five individuals with SCI/D (all cervical injuries) participated in this study, there was a mix of individuals with both complete and incomplete injuries and at least two participants with SCI/D provided feedback on each activity. Combined with feedback from clinicians, there was a minimum of five participants who commented on each activity so we feel we had sufficient data. According to Brod and colleagues [20], seven to ten interviews are typically sufficient to reach consensus. Within this study, there was a general consensus in feedback on the tool among participants with no substantial variation of opinion between groups. Although this study has demonstrated the content validity of the ABT tracking tool in the community setting, we anticipate the content validity of the tool to be similar in other settings, such as rehabilitation hospitals and the home, as the tool includes all types of ABT and their associated parameters. What may differ is the frequency with which different types of ABT and technology are used across different centers and settings, which is information that the tool will be able to track. Other limitations that may have impacted study findings are innate to qualitative research. These include interviewer [51] and social desirability bias [52]. Yet, the steps taken to establish trustworthiness and ensure rigor of this study, such as having an experienced researcher (AK) and taking reflexive notes [29], helped to minimize these sources of bias.

## Conclusion

This study evaluated the content validity of a prototype ABT tracking tool using cognitive debriefing interviews. Clinicians and people with SCI/D provided suggestions to improve the

comprehensiveness and comprehensibility of the tool. Providing the tool as an app and developing a toolkit will support the utility and implementation of the ABT tracking tool across community-based clinics in Canada.

## Supporting information

**S1 Text. Activity-Based Therapy (ABT) tracking tool: A tool that documents the details of a therapy session, such as the types of ABT practiced, the dosage and the amount of therapist assistance needed.**
(DOCX)

**S2 Text. Interview guide: The questions clinicians and people with SCI/D were asked to get their feedback on the appropriateness, comprehensiveness and comprehensibility of an ABT tracking tool.**
(DOCX)

**S1 Table. Suggested modifications to the ABT tracking tool: The revisions, with supporting verbatim quotes, suggested by clinicians and people with SCI/D to improve the ABT tracking tool.**
(DOCX)

## Acknowledgments

The authors would like to thank the participants for their time and involvement in this study.

## Author Contributions

**Conceptualization:** Anita Kaiser, Kristin E. Musselman.

**Data curation:** Anita Kaiser, Katherine Chan, Alexander Djuric, Sandi Marshall, Heather Cairns-Mills, Jennifer Leo, Kristin E. Musselman.

**Formal analysis:** Anita Kaiser, Hanan Idd, Alexander Djuric.

**Funding acquisition:** Anita Kaiser, Kristin E. Musselman.

**Investigation:** Anita Kaiser, Kristin E. Musselman.

**Methodology:** Anita Kaiser, Kristin E. Musselman.

**Project administration:** Anita Kaiser, Katherine Chan, Kristin E. Musselman.

**Supervision:** Kristin E. Musselman.

**Validation:** Anita Kaiser.

**Writing – original draft:** Anita Kaiser, Kristin E. Musselman.

**Writing – review & editing:** Hanan Idd, Katherine Chan, Alexander Djuric, Sandi Marshall, Heather Cairns-Mills, Jennifer Leo, Kristin E. Musselman.

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
