## [Decision Letter · Decision Letter 0]

24 Sep 2024

PONE-D-24-24628Validation of an activity-based therapy tracking tool for people living with spinal cord injury or disease using cognitive debriefing interviewsPLOS ONE

Dear Dr. Musselman,

Thank you for submitting your manuscript to PLOS ONE. After careful consideration, we feel that it has merit but does not fully meet PLOS ONE’s publication criteria as it currently stands. Therefore, we invite you to submit a revised version of the manuscript that addresses the points raised during the review process.

We look forward to receiving your revised manuscript.

Kind regards,

Zheng Su

Academic Editor

PLOS ONE

**Journal Requirements:**

Praxis Spinal Institute

4. In the online submission form, you indicated that Data (qualitative transcripts) cannot be shared publicly because of confidentiality. Data may be available upon request to the corresponding author and pending approval from the Research Ethics Board of the University Health Network.

5. We note that this data set consists of interview transcripts. Can you please confirm that all participants gave consent for interview transcript to be published?

If they DID provide consent for these transcripts to be published, please also confirm that the transcripts do not contain any potentially identifying information (or let us know if the participants consented to having their personal details published and made publicly available). We consider the following details to be identifying information:

- Names, nicknames, and initials

- Age more specific than round numbers

- GPS coordinates, physical addresses, IP addresses, email addresses

- Information in small sample sizes (e.g. 40 students from X class in X year at X university)

- Specific dates (e.g. visit dates, interview dates)

- ID numbers

Or, if the participants DID NOT provide consent for these transcripts to be published:

- Provide a de-identified version of the data or excerpts of interview responses

- Provide information regarding how these transcripts can be accessed by researchers who meet the criteria for access to confidential data, including:

a) the grounds for restriction

b) the name of the ethics committee, Institutional Review Board, or third-party organization that is imposing sharing restrictions on the data

c) a non-author, institutional point of contact that is able to field data access queries, in the interest of maintaining long-term data accessibility.

d) Any relevant data set names, URLs, DOIs, etc. that an independent researcher would need in order to request your minimal data set.

For further information on sharing data that contains sensitive participant information, please see: https://journals.plos.org/plosone/s/data-availability#loc-human-research-participant-data-and-other-sensitive-data

If there are ethical, legal, or third-party restrictions upon your dataset, you must provide all of the following details (https://journals.plos.org/plosone/s/data-availability#loc-acceptable-data-access-restrictions):

a) A complete description of the dataset

b) The nature of the restrictions upon the data (ethical, legal, or owned by a third party) and the reasoning behind them

c) The full name of the body imposing the restrictions upon your dataset (ethics committee, institution, data access committee, etc)

d) If the data are owned by a third party, confirmation of whether the authors received any special privileges in accessing the data that other researchers would not have

e) Direct, non-author contact information (preferably email) for the body imposing the restrictions upon the data, to which data access requests can be sent

Reviewers' comments:

Reviewer's Responses to Questions

**Comments to the Author**

1. Is the manuscript technically sound, and do the data support the conclusions?

Reviewer #1: Yes

Reviewer #2: Yes

2. Has the statistical analysis been performed appropriately and rigorously? 

Reviewer #1: Yes

Reviewer #2: Yes

3. Have the authors made all data underlying the findings in their manuscript fully available?

Reviewer #1: Yes

Reviewer #2: No

4. Is the manuscript presented in an intelligible fashion and written in standard English?

Reviewer #1: Yes

Reviewer #2: Yes

5. Review Comments to the Author

**Reviewer #1:** This study provides insight about the content validity of an activity-based therapy (ABT) tracking tool using feedback from different types of community clinicians, and people with lived experience. These are the two target groups that would use the ABT tool in the foreseeable future. Furthermore, this research provides an excellent background on not only the content of the tracking tool, but also potential implementation considerations and strategies.

The introduction and methods sections were particularly strong. The authors did an excellent job of describing previous work and demonstrating how this study was the next logical step moving towards implementation.

Major points for revision or clarification:

1. Introduction. (Line 107) The purpose is apparent within the abstract. However, within the introduction, the wording should be revised to emphasize that this is the purpose/aim in the body of the manuscript (e.g., the purpose/aim was to evaluate….)

2. Introduction. There is no mention within the introduction that this tracking tool is specific to the community setting. This is my interpretation based on the methods, please clarify whether the tool is targeted towards the community and/or other settings.

3. Methods. The background of the research team members is unclear. I suggest including these details.

4. Methods. (Line 115-118) You suggest that probing is “a technique”. If it is a “type” of cognitive debriefing interview, I suggest making this correction. If probing is a technique that is part of a cognitive debriefing interview, include a line to further define what a cognitive debriefing interview is.

5. Results. (Line 184 -193) Although your references (30,31) suggest abstraction into themes it may be more appropriate to use “categories” and “subcategories” to avoid confusion with thematic analysis or variants (Example Vears & Gillam 2022). Consider and/or clarify your position.

6. Discussion and conclusions. (Line 487-509) Was the app the only suggestion made by participants? Was this prompted?

7. Discussion and conclusions. You have described a stand-alone app to track ABT activities. Have you considered making this app part of an existing documentation app or software? This could be part of the discussion or future directions.

Minor points for revision or clarification:

1. Title. Suggest including “community” in the title.

2. References. Review the references. I noticed some included the term “(journal article)”. Also check journal abbreviations.

3. Line 162. Typographical error “of” should be “about”.

4. Line 205. Report demographic information of study dropout, if known.

5. Line 235. Place words used to describe the tool in “quotations” if they are direct quotes.

6. Line 293. Check that this quote was from a person with SCI/D, it seems like it may be a quote from a clinician.

7. Line 381. Clarify that this is the [National] SCI conference, if that is what the participant meant.

8. Line 242-245. Remind the reader which themes (categories) were deductive, and which were inductive.

**Reviewer #2:** The authors have developed a new tracking tool for Activity Based Therapy for persons with SCI/D and they outline very well the reason for creating such a tool as the current available tools leave a clinical gap. The methodology is well described and appropriate. The complete dataset is not made available by the authors as they report that they are unable to share the interview transcripts (hence the "No" to question 3 above). The results, discussion explain the discrepancies between the planned methodology and the actual sample (see below) and cover the limitations of the study well. The conclusion gives a good summation and the authors used appropriate language in stating that the results support use for "community-based" clinics since those interviewed for this project came from that background.

I did not identify any major issues in the paper.

I did identify 1 minor issue:

1. Between lines 139-143 the authors outline the proposed sample that they intended to interview based on sample size calculations. This included 10-12 clinicians and 10-12 individuals with SCI/D. The individuals with SCI/D were further planned to include an equal distribution of participants living with tetraplegia and paraplegia. In Table 2 (beginning on line 208) it shows that there were 9 clinicians interviewed (close to the expectations) but only 5 individuals with SCI/D (with all 5 having tetraplegia). It is later explained that the recruitment occurred during the COVID-19 pandemic which impacted the sample size. The authors describe in the Limitations section that their sample size is sufficient to reach consensus but fail to mention whether there is an impact from not having any individuals with paraplegia among their interviewees.

I have this listed as a minor issue as the authors do not make any claims that the tool will work for all persons with SCI/D so they keep their conclusions in line with the cohort that they were able to interview.

6. PLOS authors have the option to publish the peer review history of their article (what does this mean?). If published, this will include your full peer review and any attached files.

Reviewer #1: No

Reviewer #2: **Yes: **Shane McCullum

---

## [Author Response · Author response to Decision Letter 0]

6 Nov 2024

We thank the Reviewers and Editor for the helpful comments about our manuscript. We have addressed each comment below. Line numbers refer to lines in the tracked version of our manuscript. 

Comments from the Editor:

1. “When submitting your revision, we need you to address these additional requirements. Please ensure that your manuscript meets PLOS ONE's style requirements, including those for file naming. The PLOS ONE style templates can be found at 

https://journals.plos.org/plosone/s/file?id=ba62/PLOSOne_formatting_sample_title_authors_affiliations.pdf .”

We have reviewed PLOS ONE’s style requirements and made edits throughout the manuscript as required. 

2. “We note that the grant information you provided in the ‘Funding Information’ and ‘Financial Disclosure’ sections do not match. When you resubmit, please ensure that you provide the correct grant numbers for the awards you received for your study in the ‘Funding Information’ section.”

Apologies, but we cannot find the ‘Financial Disclosure’ section in the online system. The information provided in the ‘Funding Information’ section is correct.

3. Thank you for stating the following financial disclosure: Praxis Spinal Institute. Please state what role the funders took in the study. If the funders had no role, please state: "The funders had no role in study design, data collection and analysis, decision to publish, or preparation of the manuscript." If this statement is not correct you must amend it as needed. Please include this amended Role of Funder statement in your cover letter; we will change the online submission form on your behalf.

We have added the required statement to the cover letter: The funders had no role in study design, data collection and analysis, decision to publish, or preparation of the manuscript. We have also edited the name of the funding agency within the ‘Funding Information’ section of the online submission system – it now says ‘Praxis Spinal Cord Institute’.

4. “In the online submission form, you indicated that Data (qualitative transcripts) cannot be shared publicly because of confidentiality. Data may be available upon request to the corresponding author and pending approval from the Research Ethics Board of the University Health Network. All PLOS journals now require all data underlying the findings described in their manuscript to be freely available to other researchers, either a. In a public repository, b. Within the manuscript itself, or c. Uploaded as supplementary information. This policy applies to all data except where public deposition would breach compliance with the protocol approved by your research ethics board. If your data cannot be made publicly available for ethical or legal reasons (e.g., public availability would compromise patient privacy), please explain your reasons on resubmission and your exemption request will be escalated for approval.” 

Transcripts of qualitative interview data are considered identifiable data as it is possible to determine the individual participant given the contextual information provided during discussion. Our research ethics board does not permit sharing the qualitative transcript data – this is common practice for ethics boards across Canada and elsewhere. Hence, we are requesting an exemption. 

5. “We note that this data set consists of interview transcripts. Can you please confirm that all participants gave consent for interview transcript to be published? If they DID provide consent for these transcripts to be published, please also confirm that the transcripts do not contain any potentially identifying information (or let us know if the participants consented to having their personal details published and made publicly available). We consider the following details to be identifying information:

- Names, nicknames, and initials

- Age more specific than round numbers

- GPS coordinates, physical addresses, IP addresses, email addresses

- Information in small sample sizes (e.g. 40 students from X class in X year at X university)

- Specific dates (e.g. visit dates, interview dates)

- ID numbers

Or, if the participants DID NOT provide consent for these transcripts to be published:

- Provide a de-identified version of the data or excerpts of interview responses

- Provide information regarding how these transcripts can be accessed by researchers who meet the criteria for access to confidential data, including:

a) the grounds for restriction

b) the name of the ethics committee, Institutional Review Board, or third-party organization that is imposing sharing restrictions on the data

c) a non-author, institutional point of contact that is able to field data access queries, in the interest of maintaining long-term data accessibility.

d) Any relevant data set names, URLs, DOIs, etc. that an independent researcher would need in order to request your minimal data set. For further information on sharing data that contains sensitive participant information, please see: https://journals.plos.org/plosone/s/data-availability#loc-human-research-participant-data-and-other-sensitive-data If there are ethical, legal, or third-party restrictions upon your dataset, you must provide all of the following details (https://journals.plos.org/plosone/s/data-availability#loc-acceptable-data-access-restrictions):

a) A complete description of the dataset

b) The nature of the restrictions upon the data (ethical, legal, or owned by a third party) and the reasoning behind them

c) The full name of the body imposing the restrictions upon your dataset (ethics committee, institution, data access committee, etc)

d) If the data are owned by a third party, confirmation of whether the authors received any special privileges in accessing the data that other researchers would not have

e) Direct, non-author contact information (preferably email) for the body imposing the restrictions upon the data, to which data access requests can be sent”

Study participants did not provide permission to share transcripts as our research ethics board does not allow the sharing or distributing of qualitative transcripts. Even if names and other identifiers are removed from the transcript, transcripts are still considered identifiable due to the experiences, contexts and other details provided during qualitative interview discussions. Here are the details that you require:

a) Text transcripts of participants’ experiences using an ABT tracking tool and their feedback regarding the tool’s content.

b) Ethical. Transcripts of discussions are considered identifiable data.

c) Research Ethics Board, University Health Network.

d) Not applicable.

e) reb@uhnresearch.ca

6. “Please include captions for your Supporting Information files at the end of your manuscript, and update any in-text citations to match accordingly. Please see our Supporting Information guidelines for more information: http://journals.plos.org/plosone/s/supporting-information. “

Captions have been added. Please see lines 736-744.

We have reviewed our reference list to ensure it is complete and correct. 

Comments from Reviewer #1: 

1. “Introduction. (Line 107) The purpose is apparent within the abstract. However, within the introduction, the wording should be revised to emphasize that this is the purpose/aim in the body of the manuscript (e.g., the purpose/aim was to evaluate….)”

We have edited the text in the Introduction to ensure consistency between the aims stated in the Abstract and Introduction. Introduction, lines 107-110: As a next step in the development of the ABT tracking tool, this study aimed to evaluate the content validity of the prototype tool using cognitive debriefing interviews to determine whether the tracking tool was comprehensive, appropriate and comprehensible within the context of the construct (i.e., ABT), setting (community clinic) and population (i.e., SCI/D) of interest (20–23).

2. “Introduction. There is no mention within the introduction that this tracking tool is specific to the community setting. This is my interpretation based on the methods, please clarify whether the tool is targeted towards the community and/or other settings.”

We have added text to the study aim in the Introduction to clarify that the tool was tested in a community setting. Introduction, lines 107-110: As a next step in the development of the ABT tracking tool, this study aimed to evaluate the content validity of the prototype tool using cognitive debriefing interviews to determine whether the tracking tool was comprehensive, appropriate and comprehensible within the context of the construct (i.e., ABT), setting (community clinic) and population (i.e., SCI/D) of interest (20–23).

Although the tool was tested in a community setting, we anticipate the tool translating well to other settings. To acknowledge the limited setting of the findings, we have added the following text to the Discussion, lines 579-584: Although this study has demonstrated the content validity of the ABT tracking tool in the community setting, we anticipate the content validity of the tool to be similar in other settings, such as rehabilitation hospitals and the home, as the tool includes all types of ABT and their associated parameters. What may differ is the frequency with which different types of ABT and technology are used across different centers and settings, which is information that the tool will be able to track.

3. “Methods. The background of the research team members is unclear. I suggest including these details.”

Thank you for this suggestion. We have included this information in the Methods section. Materials and Methods, lines 163-164: The interviewer (AK who has over 25 years of experience living with SCI, ten years of experience in qualitative research and eight years of experience participating in ABT).

Materials and Methods, lines 178-180: A team member (KEM, who had extensive experience in SCI/D, ABT and qualitative research or AD who was a novice in all areas) attended each interview and took reflexive notes (29)…

Materials and Methods, lines 183-185: The interviews were audio-recorded and transcribed verbatim by a team member (HI who was a novice in all areas) in Microsoft Word (2021).

4. “Methods. (Line 115-118) You suggest that probing is “a technique”. If it is a “type” of cognitive debriefing interview, I suggest making this correction. If probing is a technique that is part of a cognitive debriefing interview, include a line to further define what a cognitive debriefing interview is.”

Thank you for this question. Yes, probing is a technique used in cognitive debriefing to determine their level of understanding and comprehension of the questions asked or items in the tool and assess how participants are interpreting each item. We have added the following sentence to further define cognitive debriefing interviews. Materials and Methods, lines 116-117: Cognitive debriefing is a technique used to assess participants’ interpretation of questions or items in a survey or other instrument during an interview.(20)

5. “Results. (Line 184 -193) Although your references (30,31) suggest abstraction into themes it may be more appropriate to use “categories” and “subcategories” to avoid confusion with thematic analysis or variants (Example Vears & Gillam 2022). Consider and/or clarify your position.”

We have adopted the terms categories and subcategories to avoid confusion. This change has been made throughout the manuscript. 

6. “Discussion and conclusions. (Line 487-509) Was the app the only suggestion made by participants? Was this prompted?”

Thank you for this question. On the Interview Guide (see Supplement 2) question 7, participants were asked “How would you prefer to track your sessions? <Probe: For example, on paper, app, another platform?>”. This question was based on previous findings from focus group interviews that queried participants on their perspective on tracking ABT.(18) In that study, participants were asked about their preferred mode of tracking and the majority of participants indicated their preference for an app, although a few participants referred to a paper-based tool as being a necessity in certain circumstances. This was similar to findings in this study, where a few clinicians acknowledged their clinic currently used paper-based tracking, yet all participants saw the benefit of and indicated a preference for digital tracking. 

7. “Discussion and conclusions. You have described a stand-alone app to track ABT activities. Have you considered making this app part of an existing documentation app or software? This could be part of the discussion or future directions.”

Thank you for this question. We are initiating a follow-up project to convert the paper-based version of the tool into an app. Part of the planning includes a review of existing exercise and rehabilitation apps to see if an existing app can be easily modified to include the ABT tracking tool. We have edited text in the Discussion to reflect the possibility that the ABT tracking tool could be incorporated into a pre-existing app. Discussion, lines 521-523: Although discontinued in 2019 pending further upgrades, the training videos are still available and may provide a useful example when developing the ABT app or modifying an existing app if deemed suitable.

Discussion, lines 564-565: Next steps will be to revise the prototype ABT tracking tool and convert it into an app or modify a pre-existing app for further feedback and psychometric tests.

8. “Title. Suggest including “community” in the title.”

Thank you for this suggestion. We have added “community setting” in the title: Content validation of an activity-based therapy tracking tool in a community setting for people living with spinal cord injury or disease using cognitive debriefing interviews

9. “References. Review the references. I noticed some included the term “(journal article)”. Also check journal abbreviations.”

Thank you for bringing this to our attention. The references have now been fixed.

10. “Line 162. Typographical error “of” should be “about”.”

Thank you for pointing this out. We have made the correction.

11. “Line 205. Report demographic information of study dropout, if known.”

We have added what is known of the participant who dropped out based on screening criteria. Materials and Methods, line 213: One clinician (female exercise physiologist) dropped out of the study shortly after enrollment due to health reasons.

12. “Line 235. Place words used to describe the tool in “quotations” if they are direct quotes.”

We have added quotations as requested.

13. “Line 293. Check that this quote was from a person with SCI/D, it seems like it may be a quote from a clinician.”

Thank you for pointing out this concern. We have verified that this quote came from participant SCI/D who made reference to likely not recording measures of exertion for several types of ABT activities.

14. “Line 381. Clarify that this is the [National] SCI conference, if that is what the participant meant.”

Thank you for this suggestion. The participant was not specific about which conference(s) they were referring to and a little before the posted quote they made reference to several conferences so we modified the quote to reflect that. Results, lines 392-397: “There’s a couple spinal cord injury conferences…I would assume that probably the spinal cord injury conference would probably be the best way to get a little bit of time in there and just say like “hey, this is what we’re doing. This is why we’re doin

---

## [Editor Report · Decision Letter 1]

26 Nov 2024

Content validation of an activity-based therapy tracking tool in a community setting for people living with spinal cord injury or disease using cognitive debriefing interviews

PONE-D-24-24628R1

Dear Dr. Musselman,

We’re pleased to inform you that your manuscript has been judged scientifically suitable for publication and will be formally accepted for publication once it meets all outstanding technical requirements.

Kind regards,

Zheng Su

Academic Editor

PLOS ONE

Additional Editor Comments (optional):

Accept
---

## [Editor Report · Acceptance letter]

6 Dec 2024

PONE-D-24-24628R1 

PLOS ONE

Dear Dr. Musselman, 

I'm pleased to inform you that your manuscript has been deemed suitable for publication in PLOS ONE. Congratulations! Your manuscript is now being handed over to our production team.

Kind regards, 

on behalf of

Dr. Zheng Su 

Academic Editor

PLOS ONE